# Digital Technology Deployment and the Circular Economy

**Martin Wynn** [1],* and **Peter Jones** [2]

1   School of Computing and Engineering, University of Gloucestershire, Cheltenham GL50 2RH, UK
2   The Business School, University of Gloucestershire, Cheltenham GL50 2RH, UK; pjones@glos.ac.uk
*   Correspondence: mwynn@glos.ac.uk

**Abstract:** The relationship between digital technologies and the circular economy, both characterised as disruptive, is attracting increasing attention in company boardrooms, policy and pressure groups and in academic communities. Nevertheless, studies to date highlight a lack of understanding of how digital technologies enable organisations to improve their resource flows and value creation to promote the circular economy. This article looks to address this gap in the academic literature by an examination of if and how a small number of organisations are using digital technologies to support their sustainability objectives and circular economy initiatives. The research approach is inductive, using questionnaires and interviews with IT professionals employed in a range of organisations. The article concludes that where organisations are pursuing circular economy initiatives, the connection with digital technology deployment is not evident, even though a more general association with sustainability is. Digital technologies are bringing about greater efficiencies, improved processes and better data management, which are supporting and enabling sustainability objectives, but a more direct linkage with the circular economy will require clearer use cases of how specific digital technologies can progress the circular economy, both within organisations and across the supply chain. Enhanced information systems that facilitate the reporting and analysis of the operational performance of circular economy activities against set objectives will also be needed.

**Keywords:** digital technologies; digital transformation; sustainability; circular economy; information systems

## 1. Introduction

In simple terms, digital technologies are electronic tools, automatic systems, and technological devices that generate, process and store information, and they are widely seen to be both disrupting and reshaping business practices and models and changing the everyday world in which we live. In essence, digital technologies allow very large volumes of data to be compressed and stored in small devices and transmitted very rapidly. These technologies are constantly evolving, but the two acronyms of SMAC (social media, mobile, analytics/Big data, cloud) and BRAID (blockchain, robotics, automation of knowledge work/artificial intelligence, internet of things and digital fabrication) are often used as umbrella terms in this context. As organisations increasingly use digital technologies in their operations, so they will have to face and address new sets of responsibilities, collectively described by Lobschat et al. [1] (p. 875) as "corporate digital responsibility", but they will also be able to seize a wide range of opportunities afforded by digitalisation. The potential transition to the circular economy (CE) is one such opportunity.

The CE has been defined by the European Union [2] (para. 1) as an "economy where the value of products, materials and resources is maintained in the economy for as long as possible, and the generation of waste minimized". The CE thus reduces both natural resource depletion and waste produced by the consumption chain [3]. The Ellen McArthur Foundation [4] (paras. 2 and 3), which was established in 2010 with the aim of accelerating the transition to the CE, argued that the "circular economy is restorative and regenerative by design" and "is based on three principles: design out waste and pollution; keep products and materials in use; regenerate natural systems". The CE concept is thus differentiated

from the traditional linear economy, in which the production process turns raw materials into waste, which, in turn, generates a range of environmental problems and leads to the loss of natural capital. It embraces all stages of the product life cycle from product design through production, marketing, consumption to waste management, recycling, and reuse. That said, the concept of the CE, and its links with the transition to a sustainable future, are complex, and further treatment can be found, for example, in Romero et al. [5], Nikolaou et al. [6] and Haseeb et al. [7].

Although the terms "digitisation", "digitalisation" and "digital transformation" are sometimes used interchangeably, there are, in fact, some differences that are worth establishing, as this may help develop an understanding of how the organisations examined in this paper are employing digital technologies to support their CE initiatives. Digitisation is simply the process of making information available and accessible in a digital format, digitalisation is the act of making processes more automated through the use of a digital format, while digital transformation is the process of devising new business applications that integrate all the digitised data and digitalised applications. It is becoming increasingly evident that there are potential linkages between the deployment of digital technologies and the transition to a CE and that they may run in parallel in a number of environments. For example, in answering the question "why is digitalization critical to creating a global circular economy", The World Economic Forum [8] (para. 1) claimed "we must accelerate the transformation to a circular economy in order to meet global climate goals by 2050", and that "this can only be achieved through focused and responsible digitalization". Okorie et al. [9] looked to provide a framework to integrate digital technologies and the circular economy, and Kottmeyer [10] (p. 17) argued that "digital technologies have the potential to close the realisation gap between theory and practice of the circular economy concept". However, Ranta et al. [11] (para. 1) suggested that, to date, work on the relationship between digital technologies and the circular economy has been based on "conceptual and review studies", which had led to "a lack of understanding of how digital technologies enable individual firms in real-life settings to improve their resource flows and value creation and capture, and thereby enable business model innovation to emerge".

This article looks to address this gap in the academic literature by an examination of if and how a small number of organisations are using digital technologies to support their CE initiatives and by looking to identify the key issues for the deployment of digital technologies in facilitating the transition to a CE. The paper includes an outline of the research methodology, a literature review of work on the relationship between digital technologies and the CE, the findings from the expert interviews that address the research questions, a discussion of some emerging issues and a conclusion that summarises the paper and outlines its main limitations and future potential research agendas.

## 2. Research Method

The research method for this study was qualitative and inductive, based on a review of the available literature and on eight in-depth interviews conducted with IT Directors or senior IT professionals in a range of organisations (Table 1). The literature review was viewed as "a means of gaining an initial impression" [12] (p. 97) that can be used "to draw the big picture" [13] (p. 1) and help develop the main research questions. As regards the interviews, the eight organisations were selected to reflect variety and pragmatism. Variety in that the organisations were from different sectors, and pragmatism in that preliminary contacts revealed that senior personnel within the organisations were prepared to participate in the interview process. This latter factor was deemed to be important in that little empirical work had been undertaken in exploring how digital technologies had been used to promote the CE. The selection of respondents was also based on personal contacts where the authors were reasonably confident the respondents would be willing to contribute and provide the required information. The anonymity of the organisation and respondent names was assured.

**Table 1.** Organisation and interviewee profiles.

| Code | Organisation Profile | Staff | Job/Role |
|------|----------------------|-------|----------|
| C1 | UK SME in Product Assembly, Sales and Marketing | 51 | IT Manager |
| C2 | German multi-national conglomerate | 160,000 | Business Process Manager |
| C3 | Global Semi-conductor manufacturer | 12,000 | Senior Programme Manager |
| C4 | UK mid-range University | 1500 | Head of Library and Information Services |
| C5 | Swiss medium-sized manufacturing industry | 700 | IT Director |
| C6 | German Health Insurance Company | 7000 | Senior Business Analyst |
| C7 | German aircraft repair and maintenance company | 20,000 | Senior IT Project Manager |
| C8 | UK Medical Products group | 720 | IT Director |

Following the initial literature review, two main research questions were set out to guide the subsequent search for relevant material and subsequently for the design of the pre-interview questionnaire (Supplementary Materials). These were:

RQ1: To what extent are organisations pursuing circular economy objectives and activities?

RQ2: Are organisations using digital technologies to support their sustainability agenda and the circular economy in particular?

Using these questions as guidelines, the authors conducted a range of Internet searches using the Google and Google Scholar search engines in the period February to May 2022. In line with Yin's [14] (p. 13) guidelines to "review previous research to develop sharper and more insightful questions about the topic", this provided an initial assessment of the key issues and provided the basis for developing a set of ten questions for the questionnaire and follow-up interview discussion.

The questionnaire was made available to interviewees prior to the interview, there being ten questions relating to organisational background, digital technology deployment and the CE in their working environment. The interviews themselves left adequate opportunity for the formulation of other questions and further enquiries. The subsequent data analysis entailed the summarizing and structuring of the data to address the research questions [12,15]. The authors believed that eight interviews were enough to allow the development of findings and to address the research questions. This is supported by Guest et al. [16] (p. 59), who "found that saturation occurred within the first twelve interviews" but that "basic elements for metathemes were present as early as six interviews". Quotations in the Findings section below are taken from the questionnaire and appended notes added in the interviews.

Given that information was provided by only one person in each organisation, these are not considered to be case studies but rather as expert interviews. It was not possible to use other personnel in these organisations to achieve a form of triangulation, but the material put forward by the participants via questionnaire and interview was discussed further subsequently by email and phone call as the data were analysed, findings were developed, and conclusions were drawn. The authors thus believe this constituted an acceptable level of validation and that the information provided was reliable. In addition, the interviewees are all known to the authors, who can vouch for the respondents in accurately reporting the situation in their organisations.

## 3. Literature Review

A growing number of authors maintain that digital technologies have a central role to play in any transition to a CE. The EIT Climate-KIC consultancy [17] (p. 4) recently

concluded that "digital solutions such as artificial intelligence, blockchain and the Internet of Things can redefine production and consumption in the 21st century, powering a new circular economy that works for people and planet alike", and Sullivan and Hussein [18] (para. 1) maintain that "innovative technology can help companies lead the transition to an inclusive, circular economy faster and more efficiently". They point out that companies such as Danone, H&M Group and DS Smith are "already leveraging these newer technologies to design waste and pollution out of their value chains while keeping products and materials in use to create positive economic, environmental, and societal impact" (para. 1). However, Kristoffersen et al. [19] (p. 241) argued that there was a dearth of analysis regarding how digital technologies can be applied "to capture the full potential of circular strategies for improving resource efficiency and productivity". The authors concluded that the "smart use of resources in the circular economy can be supported by the creation, extraction, processing, and sharing of data from digital technologies", and more forcefully that "effectively using this digital transformation will be pivotal for organizations in transitioning to, and leveraging, the CE at scale" [19] (p. 253). Okorie et al. [9] undertook a systematic literature review, and the findings revealed that while research on the CE had been on the increase, research on how digital technologies can enable a CE remains relatively limited.

Building on the work of Lucato et al. [20], Costa et al. [21] assessed six digital technologies and their impact on sustainability. They used a four-way classification of the contribution level of these technologies: Absent, Low Incorporation, Medium Incorporation and Complete Incorporation. However, the definition of these levels largely assesses the general level of deployment of the technology, and only at the Complete Incorporation level is reference made to environmental (i.e., circular economy related) issues. As an example, the definitions for the four levels of AI are shown in Table 2. The article attempts to assess the significance of the mainstream digital technologies in contributing to sustainability in three company case studies in the pulp and paper manufacturing industry. More specifically, some researchers have examined the links between digital technologies and sustainable energy production. Jose et al. [22] (p. 6), for example, conclude that "artificial intelligence associated digital technologies could increase energy efficiency to facilitate carbon trading and to realize the circular economy vision of countries to mitigate extreme weather conditions and climate change".

**Table 2.** Contribution level of artificial intelligence to sustainability in pulp and paper manufacturers (based on Costa et al. [21]).

| Contribution Level | Definition |
| --- | --- |
| Absent | Artificial Intelligence is not used in any effective capacity at our company. |
| Low Incorporation | Artificial Intelligence is integrated in the business model at peripheral activities. |
| Medium Incorporation | Artificial Intelligence is integrated in the business model at both peripheral and core activities. |
| Complete Incorporation | Artificial Intelligence is integrated in the business model at both peripheral and core activities and is positively related to environmental issues. |

Ucar et al. [23] employed a literature review and three secondary case studies to evaluate the relationship between the CE and digital technologies, principally the Internet of Things, big data analytics and AI, by integrating the principles of reuse, remanufacture and recycling. The case studies suggested that the two main roles of digital technologies were as an "enabler" and as a "trigger". In the former, digital technologies were seen to facilitate the development of the CE and improve collaboration, while in the latter, digital technologies lead to innovative processes or outcomes or associated organisational mechanisms. In this context, Owen-Jackson [24] has cited the case of Rubicon, a cloud-

based, big data platform that connects waste producers with a network of independent waste haulers and may be considered as both an enabler and a trigger. Rubicon "leverages the power of big data to enable higher diversion rates from landfill sites and find creative ways to reuse waste materials" (para. 4). It also facilitates optimised route planning for waste transport and the detailed analysis of waste data. Rather similarly, Rajput and Singh [25], in examining the connection between the CE and Industry 4.0 in the context of the supply chain, identified twenty-six significant "enabling" and fifteen "challenging" factors, which were further assessed using Principal Component Analysis (PCA).

Owen-Jackson [24] (para. 10) notes that "Internet-connected sensors can track the location, condition and availability of assets in a supply chain. Direct exchange of information via secure, decentralized channels like blockchain can keep these communications secure. Together, these innovations can optimize resources, extend lifecycles and help regenerate natural resources". More specifically, as regards blockchain, Khan et al. [26] undertook a survey of manufacturing companies in the China–Pakistan economic corridor, in which they looked to examine the role of blockchain technology in supporting the CE. Their research revealed that a number of the features of blockchain technology, notably visibility, transparency, relationship management and smart contracting, played a positive role in the CE. In a similar study, Khan et al. [27] collected data from over 200 enterprises in Malaysia, and their findings revealed that CE activities were significantly enhanced by blockchain deployment, notably in the areas of recycling and remanufacturing, circular design and circular procurement. The authors claimed that their study would help organisations achieve sustainability goals within agreed financial parameters through the integration of blockchain technology into their businesses.

A major benefit highlighted by some authors is the tracking of plastics, metals and other recyclable materials that digital technologies facilitate. In France, for example, the "French Roadmap for the Circular Economy" aims at 100% recycled plastics in the country by 2025. Renault, who are a signatory to the roadmap, note that "by recycling fabrics, reconditioning spare parts, re-using batteries from electric vehicles, and developing ever cleaner and more sustainable car-sharing services, we have put our weight behind a new and more virtuous economic model" [28] (para. 12). In this context, the traceability features of blockchain—as a secure, immutable, traceable and visible public ledger system—can reduce supply chain inefficiencies through the technology's capabilities "in tracking deficient and End of Life products, authenticating information for green products and services, measuring the carbon footprint of products, encouraging individuals to use renewable energy, and improving the effectiveness of emission trading schemes" [29] (p. 132). In a more general sense, digital technologies can engender innovation, which quickens the transition to a CE. Deloitte [30] (p. 2), in their study of the top 30 Finnish and top 50 global companies using circular business models (based on publicly available information), concluded that "digitalization enables circular economy innovation quite frequently. In some cases, digitalization may even have been the spark for creating a circular solution in the first place". Saidani et al. [31] also point out that digital tools are used to enable the measurement of circularity.

Some of the existing literature suggests that the production and deployment of digital technologies may not further the CE but may, in fact, hamper such a transition. The need for tracking and recycling digital products (e.g., phones, computers, discs, etc.) puts a significant overhead on all involved entities—consumers, producers and public authorities. Trueman [32] (para. 21), for example, in discussing Apple's AirPods product, notes that "AirPods contain tungsten, tin, tantalum, lithium, and cobalt in an unopenable plastic shell and have a lifespan of about 18 months. Like most Apple products, they're not designed to be repaired and the lithium-ion battery contained within them makes them a fire hazard in landfill sites". Even with recyclable electronic devices, adequate facilities that allow their reintroduction into the CE are often not available, and other materials such as polymers are not easily broken down or recycled. Another aspect here is the consumption of electricity and water by data centres. According to Donnelly [33], a mid-sized datacentre uses as much water as three average-sized hospitals, and this is clearly a cause for concern when

there is a global water shortage and demand for fresh water is forecast to exceed supply by 40% within the next decade.

Despite this growing research interest in the intersection of digital technologies and the CE, studies are still limited in scope, and many are based more on prediction and expectation than on researched case examples and credible analysis. Alhawari et al. [34] (p. 1) noted that although "the development of CE initiatives plays an important role in the growing digital transformation in the value chain", there had been "limited research studies in the interface of circular economy and Industry 4.0", and that "future research studies may investigate the extent to which digital transformation can increase the implementation of CE, and their influence on digital performance management". In a similar vein, Pagoropoulos et al. [35] (p. 22) reported that "the main identified gap" in their review of the extant literature was "the limited technological perspective" and that in the future, "researchers should focus on this gap, and also create more empirical results, by evaluating the application of digital technologies in actual case studies".

This review of the related literature underscores the relevance of the RQs noted in Section 2 above and provides the basis for the development of the questionnaire and interview brief. This primary data gathered from the eight expert interviews are now reported in the Findings section below.

## 4. Findings

The eight organisations in this study covered a range of industry sectors, including education, engineering services, manufacturing and health insurance (Table 1). They employ between 50 and 160,000 people and have annual turnovers of between EUR 4 million and EUR 30 billion, and as such, they provide a small but varied sample of organisations for the present study. The findings are organised below around the two research questions.

### 4.1. To What Extent Are Companies Pursuing Activities That Support the CE?

A number of activities were put forward in the questionnaire as indicative of a transition to a circular economy. These activities were in line with the conception of the CE outlined in the recent literature concerning the elimination of waste by "reusing, repairing, remanufacturing, refurbishing assets and devices, thus keeping them longer in the circle and in the loop" [32] (para. 3). These activities thus concern the reduction of waste, more efficient recycling of products, the optimisation of the product return process and the reuse of products or packaging by the end consumer (Table 3). They were collectively considered to be a reasonable representation of CE activities undertaken in organisations, and respondents were given the opportunity to comment further on the uses and benefits of digital technologies, as discussed below.

All eight organisations reported that they were undertaking some of these activities. One organisation—the mid-range UK university—reported pursuing six of the eight activities, and two others—the largest organisations in the study—were carrying out five of the eight activities. The two most supported activities were the reduction of waste or emissions and the recycling of goods and packaging within the organisation, both of which were undertaken by seven of the eight organisations. More specifically, C1 noted that "all goods packaging and scrap materials are recycled", and that "additionally, we offer free expanded polystyrene recycling for the business community within city limits". This suggests that the assertion of Sullivan and Hussein [18] (para. 1) that "ten years from now there will be no tolerance of waste in the value chain", may well be realistic. It was also significant that those activities that required collaboration with business partners, end-users or customers were pursued by only one or two of the organisations studied.

**Table 3.** Circular economy activities in the organisations studied.

| | Circular Economy Activities/Respondent | C1 | C2 | C3 | C4 | C5 | C6 | C7 | C8 |
|---|---|---|---|---|---|---|---|---|---|
| 1. | Reduction of waste and/or emissions within the organisation | Yes | Yes | Yes | Yes | Yes | | Yes | Yes |
| 2. | Production or purchase of readily recyclable products | | Yes | | Yes | | | Yes | |
| 3. | Production or purchase of products with a high proportion of recycled material | | Yes | | Yes | | Yes | Yes | |
| 4. | Optimisation of the product returns processes | Yes | Yes | Yes | Yes | Yes | | | |
| 5. | Recycling of goods and packaging within the organisation | Yes | Yes | Yes | Yes | Yes | | Yes | Yes |
| 6. | Encourage the reuse of product and/or packaging by the end consumer | | | | Yes | | | | |
| 7. | Cooperation and collaboration with partners in the transition to a circular economy | Yes | | | | | | | Yes |
| 8. | Adapting the organisation's business model | | | | | | | Yes | Yes |
| 9. | Other | | | | | | Yes | | Yes |

There appeared to be no obvious correlation between the size or type of the organisations and the number of CE activities being pursued. For example, CI, the smallest of the eight organisations with just 51 staff, are pursuing more CE activities than the German semi-conductor manufacturer, with 12,000 staff. In addition to the CE activities put forward in the questionnaire, one organisation, the UK medical products group (C8), added that the "company established an Environmental, Social and Governance (ESG) Committee in 2021 to start to formalise and report on actions we already take and develop a stronger set of strategic objectives. This information will be included in future reports and accounts as provided to investors". The German healthcare assurance company (C6) also added that "adjustments of printers to print only duplex and print only 'relevant' documents" was also viewed as supporting the CE.

*4.2. How Are Organisations Currently Using Digital Technologies to Support Their Sustainability Agenda and the Circular Economy in Particular?*

As regards the technologies deployed, all nine of the SMAC/BRAID technologies were considered to be of relevance in advancing sustainability or the CE in at least some of the organisations studied. The organisations reported that a range of digital technologies was supporting a more circular economy and the transition to greater sustainability, but this linkage varied between organisations. Cloud, mobile, AI and analytics/big data emerge as those technologies most linked to sustainability and the CE, closely followed by IoT, robotics and digital fabrication (Table 4). The larger organisations are using a wider range of these technologies in support of sustainability or CE objectives, reflecting their larger IT budgets and resources.

Table 4. Questionnaire responses: are these technologies used in support of sustainability and/or circular economy objectives?

| Digital Technology/Interviewee | C1 | C2 | C3 | C4 | C5 | C6 | C7 | C8 |
|---|---|---|---|---|---|---|---|---|
| Social Media | | | | Yes | | | Yes | |
| Mobile Computing/Apps | Yes | Yes | Yes | Yes | | Yes | Yes | |
| Analytics and Big Data | | Yes | Yes | Yes | | Yes | Yes | |
| Cloud Computing | Yes | Yes | Yes | Yes | Yes | | Yes | Yes |
| Blockchain | | | | Yes | | | Yes | |
| Robotics | | Yes | Yes | | | Yes | Yes | |
| Artificial Intelligence/Automation of Knowledge Work | Yes | Yes | Yes | Yes | | Yes | Yes | |
| Internet of Things | | Yes | Yes | Yes | | | Yes | |
| Digital Fabrication/Digital Twin/3-D Printing | | Yes | Yes | | Yes | | Yes | |

*Social Media*, particularly Facebook and Instagram, were widely used in the organisations studied for marketing and other purposes; but more specifically, social media was used by two of the organisations to support the circular economy. C7 saw it as a means of "improving communications" and noted "the topic of the environment is becoming more and more important, and social media can provide support here". C4, on the other hand, saw social media as a means of "regular promotion of positive initiatives to share knowledge", including in support of CE activities.

A wide range of *Mobile Computing Applications* was used to support mobile teams at customer locations, enhance business productivity, for scanning, the processing and payment of bills and operate remote control devices. Six organisations reported using mobile apps to promote sustainability objectives. C7 noted that in the aircraft maintenance business, this improved efficiencies by "identifying which product is where in the Product Life Cycle", and C3, working in a manufacturing company, similarly noted that mobile computing "improved shop floor staff efficiency as they can faster access relevant data in case of maintenance issues or similar". C2 saw "consumption measurement" as a major benefit in the multi-national manufacturing conglomerate, and C1 considered "paper reduction and fuel economies" as benefits derived from mobile apps. In the healthcare insurance industry, C6 noted "faster payments, reducing post processes and costs" as benefits, whilst in the university environment, C4—whilst adopting a wider definition of mobile computing—observed a number of efficiency gains, including the "switch to Teams telephony reducing the need for separate phone handsets", the "switch to laptops reducing duplication of staff having multiple computers", and "home working reducing travel".

Five respondents reported that *Analytics and Big Data* were supporting sustainability objectives. C7 noted that the aircraft maintenance company was benefitting from increased efficiencies through the use of the "AVIATAR platform that allows airlines to decide who can access their operational flight and technical data". C3, working in the semi-conductor business, saw analytics and big data as helping to save non-conformity costs—for example, "to find new correlations between product failures", whilst C4 saw "work to integrate Building Management Systems into cloud-based analytics systems [as] enabling better oversight and management of all built assets". More specifically, C2 reported that advanced analytics (using the Microsoft Power BI software) was "used for monitoring and controlling sustainability KPIs"; and C6 saw "gaining valuable information at a glance" as a significant benefit that generally supported sustainability. As regards *Cloud Computing*, C7 noted the value of the "Cloud platform to optimize flight operations", and C5 saw benefits in reduced energy consumption. C3 again saw value in saving non-conformity costs, an example being "to provide a high-performance infrastructure for data analysis". C4 noted that the "move to Cloud" was "reducing the need for as many physical assets on premise". C8 reported that "environmental reporting on materials used, waste created and energy consumption"

benefited from the use of cloud-based systems, and C1 viewed cloud computing as a supporting technology for "paper reduction and fuel economies".

Only one organisation saw a link between the use of *Blockchain* and sustainability, which contrasts with the findings of Khan et al. [26,27] and Erol et al. [29] noted above. C7 reported that the aircraft maintenance company were using "blockchain technology to track, trace and record aircraft parts in the global supply and distribution chains". However, C4, in the university environment, were "considering the use of blockchain to create an auditable record of achievement, enabling employers to quickly validate student awards".

*Artificial Intelligence* is used by six of the organisations in support of sustainability or CE objectives. C7 noted that "artificial intelligence helps to utilize the huge volumes of data generated from diagnostics", and C6 similarly observed that through AI applications, they were "building up knowledge and implementing efficient information processes". For C2, "artificial intelligence is used to increase efficiency, minimise scrap and optimise resources", and C1 similarly saw AI as "enabling fuel economies". C4, working in a university environment, saw "opportunities to introduce RPA [Robotic Process Automation] into specific areas of the organisation to enable efficient workflows". C3 saw AI as contributing to saving non-conformity costs, noting "we apply machine learning methods to identify process anomalies and unknown failure causes". More specifically, as regards *Robotics*, C7 reported that for aircraft maintenance, they deployed a "stationary robot for intricate inspection and lasting repair. The robotic can detect and repair large damaged areas on the wings and the fuselage of an aircraft—on site and even upside down". C3 reported that robotics were used in the semi-conductor industry "to automate material handling processes in order to reduce the risk of manual failures to save non-conformity costs". C6 noted that chatbots were used for "efficient processes and payments" in the healthcare insurance company, and C2 highlighted the use of collaborative robots ("cobots"), whereby humans and robots work and interact in close proximity.

As regards the *Internet of Things*, a range of linkages with sustainability was voiced. C4 commented that the "introduction of movement sensors to manage light and inform room usage will feed overall space planning needs, and should reduce needs where rooms are booked but are not utilised". C2 noted that IoT "sensors for improving processes, supervised and unsupervised machine learning" were being introduced, and C7 saw IoT as a "platform that supports facility monitoring, automation and data-driven decision making". C3 reported that IoT was used "to integrate production machines with enterprise IT systems, which allows remote control and checking of various start conditions in order to reduce the risk of process failures".

The use of *Digital Fabrication/Digital Twin/3-D Printing* in support of sustainability and CE objectives was evident only in the four manufacturing companies. C7, for example, noted the benefits of using "3D printing for maintenance of aircraft parts," and C2 observed, "3D improves sourcing and reduces transport—digital twin improves maintenance and reduces inventory of spare parts". C5 affirmed that "inhouse prototyping reduce mainly transportation costs", and C3 summarised: "we collect extensive amounts of data per production machine (sensor data, logistics data, maintenance data etc.) and have detailed master data structures for those machines. In combination, this allows the creation of digital twins, which helps to gain new insights regarding unclear machine behaviours based on 'what-if' scenarios".

## 5. Discussion

The findings highlight a number of issues that merit further discussion. First, most of the organisations studied are pursuing CE activities that are contained within the bounds of their own organisations, but very few CE activities are in evidence that extend outside the organisation—with customers or partners in the supply chain. The first five activities listed in Table 3, which concern internal operations, are undertaken by the majority of organisations (65%), whilst there was very limited involvement (19%) in activities 6 and 7, which concern cross-supply chain operations. At the same time, only two of the eight

organisations are witnessing any adaptation of their business model to accommodate CE thinking, values or operations. This suggests that the focus to date has mainly been inward-looking and that cross-supply chain opportunities for advancing the CE are only just starting to be explored. C1, for example, noted that there were "ongoing discussions within the industry and regulatory bodies and codes of practice pertinent to this industry. This is at an early stage where no practical impact has yet been achieved". C4 also noted that, as regards cross-supply chain collaboration, "we need to give more thought to this given our commitment to sustainability". The IT manager at C1 added, "there is a growing trend within public sector contracts for increased mention of relevant circular economy concepts. Contracts are renewed/bid for on a one to three-year cycle so evidence is slowly being seen. At this time, it is clear that these concepts are not critical factors in bid scoring". Equally, however, the limited involvement in pursuing CE activities might also be seen to reflect the need to incorporate digital transformation more fully into all areas of the company's activities, as outlined earlier.

Further, as regards the optimisation of product returns, C1—working in the product assembly sector—commented, "this is currently under significant discussion within the industry particularly in regard to battery waste within the European Union and 'return to manufacturer'. No clear process or consensus has been reached within the industry as yet". Indeed, research studies on how digital technologies enable and support the CE within the extended supply chain are few and far between [9,36], with very little systematic or sector-specific research in this field [37]. It is clear, nevertheless, that progressing CE operations across the supply chain involves coordinated actions by many stakeholders that may entail significant changes in how products are designed, used and managed at end-of-life [38]. Although this is a growing area of research and practice [39], the organisations studied here are mainly focused on in-house operations at present.

Second, the organisations studied evidenced a clear link between the deployment of digital technologies and sustainability, but the link to the CE was not so strong. Only one of the CE activities was seen to be directly supported by digital technology deployment—the multi-national conglomerate (C2) using AI to support waste reduction. Nevertheless, when asked about the use of individual technologies, further examples of sustainability (if not CE) benefits emerged (Table 5). These included the reduction of energy consumption, reduction of transportation costs, greater knowledge and awareness and improved sustainability reporting. However, the main benefits were seen as generating improvements in systems, data management and processes and in a range of other efficiency gains. Although these benefits are no different in principle from those used to support the argument for IT investment in general for the past 30 years or more, the scale and impact of such gains, and their combined transformational impact, are arguably perceived to be on a different scale. This aligns with Mapfre [40] (para. 8), who concluded that "new technologies will be what makes the paradigm shift possible, as they provide the tools that are able to lower costs, automate tasks, and even create economic value".

Third, core information systems are evolving to include enhancements that allow users to record, measure and report on progress towards the CE. Jones and Wynn [41,42] pointed out how Enterprise Resource Planning (ERP) software systems were being enhanced to include functionality to measure and report on aspects of sustainability in the mining and hospitality industries. This has since become a cross-industry issue, with the major ERP vendors adding new functionality to their software products to accommodate sustainability issues and provide connectivity with digital devices and technologies. As O'Donnell [43] (para. 4) has observed, "because ERP systems are at the heart of many of the processes and systems involved in the manufacturing and distribution of products, they can play a critical role in enabling companies to move to the circular economy". More specifically, SAP, the leading ERP software vendor, has highlighted four key processes central to a circular business model, which have the potential benefits of "minimising costs, increasing customer satisfaction, mitigating risk, growing profits, and enabling resilience" [44] (para. 5). These are: Responsible Sourcing, Responsible Production, Responsible Consumption and

Responsible Recovery and Reuse. The company maintains that integrated information systems with the requisite functionality are required to underpin a transition to the circular economy in these areas. The need for information again becomes centre stage—it must be recorded, analysed and made available for informed decisions to be taken regarding CE operations. This will lead to the development and deployment of a new generation of integrated information systems—in the same way that the problems of inconsistent reporting from sales and financial systems in the late 1980s and early 1990s led to the widespread adoption of the early ERP systems.

**Table 5.** Sustainability-related benefits from digital technology deployment.

| Sustainability Benefits/Technology Used | Social Media | Mobile Apps | Analytics/ Big Data | Cloud | Blockchain | Robotics | AI/KW Auto | IoT | Digital Fab |
|---|---|---|---|---|---|---|---|---|---|
| 1. Waste reduction | | | | | | | C2 | | |
| 2. Reduction of energy consumption | | | | C5 | | | | | |
| 3. Reduction of transportation costs | | | | | | | | | C2 C5 |
| 4. Augment CE knowledge and awareness | C7 C4 | | C6 | | | | | | |
| 5. Sustainability monitoring | | | C2 | C8 | | | | | |
| 6. Systems, data management, and process improvements | | | C4 | C3 | C7 | C3 C6 C7 | C3 C6 C7 | C3 C2 C4 C7 | C3 C7 |
| 7. Other efficiency gains | | C7 C3 C2 C6 C4 | C7 C3 | C7 C4 | | | | | C2 |

Fourth, the IT function can play a key role in advancing the CE, both in terms of its own operations and in being the catalyst and exemplar for the organisation as a whole. Turbonomic [45] (p. 1) recently observed that in the context of environmental impact and sustainability, "IT has the opportunity to reduce impact immediately, through optimization of cloud and data centre resource consumption and minimize it continuously. Prioritizing sustainable resource consumption offers quick wins that last, as you also initiate longer-term investments in renewable energy, more efficient hardware, and the like". More specifically, they note that in terms of emissions, "organizations can implement a circular economy strategy. This strategy aims to reduce carbon emissions and overall environmental impact by using recyclable materials, refurbishing hardware to extend the life of assets, and responsible disposal of environmentally sensitive materials" (p. 3). Not only can the IT function play a lead role in this, but they can also facilitate the demonstration and piloting of new information systems, such as those discussed above, that provide new functionality to support sustainability monitoring and CE activities.

## 6. Conclusions

This article set out to explore two main issues—the degree to which organisations are pursuing activities linked to a transition to the CE and the role of digital technologies in this endeavour. There are clearly limitations to the research in that it relies on evidence from IT professionals in just eight organisations based in Europe, and only one person in each of the organisations provided the information reported here. Ideally, a more rounded view could have been obtained if other personnel in these organisations had been interviewed as well. Nevertheless, the authors believe that the evidence from these eight respondents supports the view that in Western Europe, organisations are, or at least claim to be, pursuing a range of activities linked to the CE. Equally, most organisations are using some digital technologies to support their operations and core business processes. The link between the two activities, however, is more tenuous. Certainly, sustainability is seen to be connected to digital technology deployment, but a clear link between CE activities and the use of digital technologies was not evident in the organisations studied.

This conclusion is somewhat at odds with some of the reviewed literature that suggests a clearer connection between digital technology deployment and the CE. This may be partly explained by the difference between commentaries on what is feasible in the future (as in the literature) and what is grounded in reality and reflects the current situation (as in the organisations studied here). It may also be that there is no major link in practice between the type of CE activity discussed here and digital technologies. Rather, there is a link between IT deployment in general and broad sustainability benefits, but the more specific connection between digital technologies and the CE is not yet proven or demonstrably seen in practice. This is notably the case with blockchain technology, which is discussed in some of the reviewed literature as a key element in the future transition to the CE. However, in the eight organisations studied here, there was little appreciation of this linkage and minimal use of this technology in practice. This is perhaps because, as Cagno et al. [46] concluded, analyses of the linkages between digital technologies and the circular economy remain relatively undeveloped, and there is thus a need for further empirical research of specific circular economy practices.

This points to some clear avenues for future investigation. First, the set of circular economy activities put forward in the questionnaire and interviews with respondents needs reviewing, expanding and refining as necessary. Bresanelli et al. [47], for example, identified eight specific functionalities—improving product design, attracting target customers, monitoring and tracking product activity, providing technical support, providing preventive and predictive maintenance, optimizing the product usage, upgrading the product, enhancing renovation and end-of-life activities—all of which were seen to be important in the transition to a CE. A clearer and more commonly agreed set of activities and supporting competencies would provide a sounder basis for future research.

Secondly, detailed case studies of the role digital technologies are playing in support of such activities would be of value not only within organisations but also across the supply chain. Cagno et al. [46] similarly called for the adoption of a case study methodology to develop a deeper understanding of the relationships between digital technologies and the circular economy. Such cases could be seen as part of what is sometimes termed "circular supply chain management" (CSCM), an emerging field of research that potentially encompasses innovation in business models and all functions across the supply chain [48].

Thirdly, empirical research could usefully explore the role of different facets of organisations in promoting the role of digital technologies and the transition to the CE. Chauhan et al. [49], for example, have suggested that enquiries might involve a focus on the digital culture of organisations to explore the role of organisational pressure in promoting links between digital technologies and the CE. Such research might also profitably focus on how a range of managers and executives within organisations develop their understanding of the role of digital technologies in promoting the CE.

Notwithstanding such new research initiatives, it is clear that a transition to the CE will be neither easy nor straightforward. Mapfre [40] (paras. 3 and 4) point out that such a transition "involves changing the way things have been done for decades", and suggests that "the main stumbling block of which will be changing the mentality of millions of consumers around the world, especially in countries with higher economic levels". Sullivan and Hussain [18] (para. 17) also maintain that "technology will not 'fix' our ecological crisis", but nevertheless conclude that "technology has incalculable potential to enable humanity to be the best stewards of the biosphere, and usher into existence a truly inclusive circular economy faster, more effectively, and more efficiently to create positive economic, environmental, and societal impact". It is to be hoped this more positive assessment endures, and that digital technology will indeed play an effective part in engendering the transition to a circular economy.

**Supplementary Materials:** The following are available online at https://www.mdpi.com/article/10.3390/su14159077/s1, Questionnaire.

**Author Contributions:** Conceptualisation, M.W. and P.J.; methodology, M.W. and P.J.; primary research, M.W.; secondary research, M.W. and P.J.; formal analysis, M.W. and P.J.; writing—original draft preparation, M.W.; writing—review and editing, P.J.; supervision, M.W.; project administration, M.W. All authors have read and agreed to the published version of the manuscript.

**Funding:** This research received no external funding.

**Institutional Review Board Statement:** Not applicable.

**Informed Consent Statement:** Not applicable.

**Data Availability Statement:** The original data sources are not available publicly because of assurances given regarding confidentiality and anonymity.

**Acknowledgments:** Grateful acknowledgement is given to the eight IT professionals who contributed to this article by providing information on the subject matter in their organisations.

**Conflicts of Interest:** The authors declare no conflict of interest.

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
