# Peer review of "Digital Technology Deployment and the Circular Economy"

_sustainability, doi:10.3390/su14159077_

Round 1

Reviewer 1 Report

The article discusses a very interesting and current topic. To help improve the article, below the authors can find some recommendations/suggestions:

- It is my belief that the richness of this article lies on its data. However, I would also recommends applying the principle of chain of evidence i.e. use several sources of data collection for triangulation and corroboration. To add more sources of data collection may be relevant given that the focus is mostly on the interviews.

- Another positive aspect of the article is related to the fact that the authors refer to theoretical saturation. However, it seems that the issue of validity and reliability was left out. Can you add further discussion in this regard?

- With regard to elementary concepts, I think it would be useful to make a distinction between digitization-digitization-digital transformation. These terms are used differently today. The clarification of these concepts may benefit the article. Below are some reading suggestions:

doi.org/10.1007/978-3-319-77703-0_41

doi.org/10.1007/978-3-030-43616-2_47

- I also recommend further reading on the circular economy, which was recently published in the journal Sustainability (MDPI).

doi.org/10.3390/su13063299

Overall, the article presents some interesting theoretical and management contributions. In my opinion, its publication will further enlighten the field of study.

Author Response

Please see attached file for Reviewer 1 comments

Reviewer 2 Report

Like your paper - well structure.

My concern with the analysis is that impact of the differences between the eight interviewed organisations are not used to explain differences in data. Include this in your analysis (text) as well as in the tables - note tables are very insightful.

Furthermore, it would be good to include the questionnaire in the appendix - in the suggestions for further research there can be made specific references to questions/sections of the questionnaire. 

Secondly the limitations of this paper should include more strongly the limitations of the number of interviews and that there is only one interview per organisation - expert interviews versus case study - make clear you have conducted expert interviews - despite the reference to case studies (Yin).

Author Response

Like your paper - well structured.

My concern with the analysis is that impact of the differences between the eight interviewed organisations are not used to explain differences in data. Include this in your analysis (text) as well as in the tables - note tables are very insightful.

RESPONSE: We have added a paragraph on this aspect in section 4.1.

Furthermore, it would be good to include the questionnaire in the appendix - in the suggestions for further research there can be made specific references to questions/sections of the questionnaire. 

RESPONSE: We can add the questionnaire, although our personal view is that this does not add much to the article. The editor can decide if it should be included. We happy either way.

Secondly the limitations of this paper should include more strongly the limitations of the number of interviews and that there is only one interview per organisation - expert interviews versus case study - make clear you have conducted expert interviews - despite the reference to case studies (Yin).

RESPONSE: We have emphasised that these are expert interviews rather than case studies, and also noted the limitations of the study in the Conclusions.

Reviewer 3 Report

The study presented in the article does not have shows significant novelty in the methodology; hence the authors need to change the methodology and approach to address the issue. Then it will be considered for possible publication. 
Also, the authors have to address the following comments. 

1. The articles require an extensive English revision. Use the software for the grammar check. 

2. How could the conclusions be drawn based on the eight interviews only? 

3. There are only two research questions to achieve the said objectives. Restructure it appropriately and increase the number of research questions. 

4. Literature review section should be above the research method section. '

5. The methodology used in the article (analysis by interviews) has less impact. Since the interviewees' state of mind, emotional quotient, and experience affect the survey outcomes. Hence, change the methodology and used the techniques such as AI and machine learning to get precise results. 

Kindly refer to the following articles but are not limited to these only for the methodology and article structuring:

a. Wilson, M.Paschen, J. and Pitt, L. (2022), "The circular economy meets artificial intelligence (AI): understanding the opportunities of AI for reverse logistics", Management of Environmental Quality, Vol. 33 No. 1, pp. 9-25. https://doi.org/10.1108/MEQ-10-2020-0222

b. Jose, R., Panigrahi, S.K., Patil, R.A. et al. Artificial Intelligence-Driven Circular Economy as a Key Enabler for Sustainable Energy Management. Mater Circ Econ 2, 8 (2020). https://doi.org/10.1007/s42824-020-00009-9

c. Shubhangini Rajput, Surya Prakash Singh, 'Connecting circular economy and industry 4.0', International Journal of Information Management, Volume 49, 2019, Pages 98-113, ISSN 0268-4012, https://doi.org/10.1016/j.ijinfomgt.2019.03.002.

d. S. Sayyad, S. Kumar, A. Bongale, P. Kamat, S. Patil and K. Kotecha, "Data-Driven Remaining Useful Life Estimation for Milling Process: Sensors, Algorithms, Datasets, and Future Directions," in IEEE Access, vol. 9, pp. 110255-110286, 2021, doi: 10.1109/ACCESS.2021.3101284.

Refer to the above articles for the methodology and article structure and update the article accordingly. 

Author Response

Please see attached file for Reviewer 3 responses

Reviewer 4 Report

Congratulations to the authors of the (very interesting) research problem, and at the same time I appreciate the effort put into preparing the article.

As a reviewer, however, I would like to make a few comments:

1. The methodology does not explain why these companies were selected for the research? Besides, despite the explanations of the authors, in my opinion the interviews with 8 entities may not be reliable.

2. In my opinion, the "Literature review" part is a bit chaotic. It is not (or not quite clearly) indicated why these circular economy activities were chosen (Table 3). It is similar with the highlighted digital technologies. What was the literary basis for this choice? The technologies listed in Table 4 are Industry 4.0 tools, but "Social media" does not stand out among them. What is the purpose of the detailed description of "Contribution level of artificial intelligence to sustainability in pulp and paper manufacturers"?

I also think that some of the arguments in this section (line 180-228) would be more appropriate for the “Discussion" or "Concslusion" sections.

3. The research carried out is interesting, but in my opinion too general to be treated as a scientific contribution.

Despite my criticism, I encourage the authors to improve the article and resubmit it.

Author Response

Please see attached file for Reviewer 4 responses.

Round 2

Reviewer 2 Report

Thank you for processing the feedback - I'm happy with the updated version.

Author Response

Thank you for your positive review.

Reviewer 4 Report

Dear Authors,

I accepted your improvements and explainations.

Please consider enriching your bibliography with the following publications::

Haseeb, M., Hussain, H. I., Kot, S., Androniceanu, A., & Jermsittiparsert, K. (2019). Role of social and technological challenges in achieving a sustainable competitive advantage and sustainable business performance. Sustainability11(14), 3811.

Afonasova, M. A., Panfilova, E. E., Galichkina, M. A., & Åšlusarczyk, B. (2019). Digitalization in economy and innovation: The effect on social and economic processes. Polish Journal of Management Studies, 19 (2): p. 22-32

Author Response

Thank you - We have included one of your suggested references (Haseeb et al.) in the article.